# Identification of a Novel Aflatoxin B_1_-Degrading Strain, *Bacillus halotolerans* DDC-4, and Its Response Mechanisms to Aflatoxin B_1_

**DOI:** 10.3390/toxins16060256

**Published:** 2024-05-31

**Authors:** Jia Guo, Hanlu Zhang, Yixuan Zhao, Xiaoxu Hao, Yu Liu, Suhong Li, Rina Wu

**Affiliations:** 1College of Food Science, Shenyang Agricultural University, Shenyang 110866, China; echoguo@syau.edu.cn (J.G.); zhaoyixuan9802@163.com (Y.Z.); 2021220129@stu.syau.edu.cn (X.H.); 2022240245@stu.syau.edu.cn (Y.L.); 2Engineering Research Center of Food Fermentation Technology, Liaoning, Key Laboratory of Microbial Fermentation Technology Innovation, Shenyang 110866, China; 3Greens SCI. & TECH. Development Co., Ltd., Tangshan 063299, China; zhanghanlu20240527@163.com

**Keywords:** aflatoxin B_1_, *Bacillus halotolerans*, biodegradation, thermostable extracellular proteins, response mechanisms

## Abstract

Aflatoxin B_1_ (AFB_1_) contamination is a food safety issue threatening human health globally. Biodegradation is an effective method for overcoming this problem, and many microorganisms have been identified as AFB_1_-degrading strains. However, the response mechanisms of these microbes to AFB_1_ remain unclear. More degrading enzymes, especially of new types, need to be discovered. In this study, a novel AFB_1_-degrading strain, DDC-4, was isolated using coumarin as the sole carbon source. This strain was identified as *Bacillus halotolerans* through physiological, biochemical, and molecular methods. The strain’s degradation activity was predominantly attributable to thermostable extracellular proteins (degradation rate remained approximately 80% at 90 °C) and was augmented by Cu^2+^ (95.45% AFB_1_ was degraded at 48 h). Alpha/beta hydrolase (arylesterase) was selected as candidate AFB_1_-degrading enzymes for the first time as a gene encoding this enzyme was highly expressed in the presence of AFB_1_. Moreover, AFB_1_ inhibited many genes involved in the nucleotide synthesis of strain DDC-4, which is possibly the partial molecular mechanism of AFB_1_’s toxicity to microorganisms. To survive under this stress, sporulation-related genes were induced in the strain. Altogether, our study identified a novel AFB_1_-degrading strain and explained its response mechanisms to AFB_1_, thereby providing new insights for AFB_1_ biodegradation.

## 1. Introduction

Aflatoxins are a group of noxious difuran coumarin derivatives and are mainly produced by the *Aspergillus* species (e.g., *As. flavus* and *As. parasiticus*) [1]. They primarily spread through contamination of various foodstuffs (e.g., nuts, corn, and oil by-products) during crop growth, harvest, and storage [2,3,4]. Moreover, this toxin can barely be degraded naturally because of its high stability. Thus, approximately five billion people are at the risk of chronic exposure to aflatoxin worldwide [5]. Among the identified aflatoxins, aflatoxin B_1_ (AFB_1_) is regarded the most toxic, carcinogenic, and mutagenic because of the C8-C9 double bond of the difuran ring and the lactone ring within the coumarin ring [6].

In the last decade, several physical, chemical, and biological approaches have been reported for AFB_1_ degradation [5]. Compared with other methods, biodegradation is the most promising alternative because of its high specificity, eco-friendliness, and harmlessness to nutritional and organoleptic properties of food [6]. Until now, many AFB_1_-degrading strains have been identified, such as *Bacillus subtilis* UTBSP1 [7], *Pseudomonas putida* [8], *Mycobacterium smegmatis* mc^2^ 155 [9], *Rhodococcus pyridinivorans* [10], *As. niger* FS10 [11], *Zygosaccharomyces rouxii* [12], *Armillariella tabescens* [13], and *Trametes versicolor* [14]. However, the response mechanisms related to AFB_1_ toxicity, degradation, and adaptation in degrading strains remain unknown.

Degrading strains chiefly mediate AFB_1_ degradation by producing enzymes that convert this toxin into less toxic or nontoxic metabolites. Most of the reported degrading enzymes are oxidoreductases, including oxidase (e.g., aflatoxin oxidase enzyme, AFO [13] and laccases [15]), peroxidase (e.g., manganese peroxidase, MnP) [16], and reductases (e.g., F_420_/H_2_-dependent reductases) [9]. The degradation mechanisms of oxidase and peroxidase to AFB_1_ are mainly oxidation and hydroxylation reactions. The major chemically active location for these reactions is the difuran ring due to the presence of a double bond in conjugation with an oxygen atom [17]. AFO from *Ar. tabescens* and MnPs from the white-rot fungus such as *Phanerochaete sordida* YK-624 could oxidize the furan ring of AFB_1_ to 8,9-epoxide formation, further forming 8,9-dihydrodiol through hydrolysis [16,18]. A conversion of AFB_1_ to AFQ_1_ is also a common degradation pathway, which was found in laccase of Lac2 from the white-rot fungus [19], CotA laccase from *B. licheniformis* [20], and dye-decolorizing peroxidase type B [21]. The major targets of reductases are unsaturations in furan and lactone rings and the α-β unsaturated carbonyl group [17]. F_420_/H_2_-dependent reductases identified from *M. smegmatis* could reduce α,β-unsaturated esters of AFB_1_ [9].

As determined by the structures of AFB_1_ and degradation products, hydrolysis, demethylation, demethoxylation, and decarbonylation reactions are also involved in the degradation mechanisms [17]. In many AFB_1_-degrading strains, hydrolysis of the lactone ring has been reported as a starting point [17]. After hydrolysis, the presence of the α-β unsaturated in the product increases its chemical activity, leading to a series of degradation reactions, including decarboxylation and the cleavage of the cyclopentenone ring, which convert AFB_1_ to AFD_1_ and further to AFD_2_ [8]. Moreover, some demethylated, demethoxylated, and decarbonylated products were also found in the biodegraded products of *T. versicolor* [14] and *Tetragenococcus halophilus* CGMCC 3792 [22]. However, the enzymes involved in the above reactions have not been identified. Therefore, more AFB_1_-degrading enzymes, especially of new types, need to be identified. This will contribute to efficient AFB_1_ degradation through genetic engineering methods.

With the advancements of high-throughput sequencing and bioinformatics, omics technologies offer a new in-depth insight into the response mechanism, which will help identify more genes encoding degrading enzymes [23]. Xu et al. identified a gene encoding the novel zearalenone degradation-associated thioesterase from *B. amyloliquefaciens* H6 through transcriptomic analysis [24]. On investigating the detoxification mechanism of *R. pyridinivorans* GF3 in response to thraquinone-2-sulfonate (ASA-2) through transcriptomic analysis, Wang et al. found that cytochrome P450 and short-chain dehydrogenase/reductase are involved in ASA-2 degradation [25]. By performing transcriptomic analysis, Wei et al. explored the response mechanism of *Cryptococcus podzolicus* Y3 under ochratoxin A stress [26]. Protein processing in *C. podzolicus* Y3 was inhibited by ochratoxin A, and *C. podzolicus* Y3 improved the excision repair pathway to protect genetic information.

In this study, 12 AFB_1_-degrading strains were screened from moldy maize, moldy rice, and strains stored in our laboratory (isolated from Chinese traditional fermented foods). Among them, strain DDC-4 exhibited the highest degradation activity and was identified as *B. halotolerans* through physiological, biochemical, and molecular methods. The active component and its characteristics were explored, and transcriptomic analysis was performed to explore the response mechanisms of strain DDC-4 to AFB_1_. Several candidate AFB_1_-degrading genes, especially the previously ignored alpha/beta hydrolase (arylesterase) gene, were mined.

## 2. Results

### 2.1. Isolation and Identification of AFB_1_-Degrading Strains

The modified Hormisch medium, containing coumarin as the sole carbon source, was used for obtaining potential AFB_1_-degrading strains [27]. Coumarin is the basic molecular structure of aflatoxin B_1_ with lower price and more secure. The strains that could grow on modified Hormisch medium have the ability to utilize coumarin as their carbon source indicating they might also be able to degrade aflatoxin B_1_ [28]. In total, 12 strains were isolated from various sources, namely 7 strains from moldy rice (i.e., ZYX1–ZYX7, respectively), 3 strains from moldy maize (i.e., DC-1, DC-3, and DC-5, respectively), and 2 strains from among those stored in our lab (i.e., DDC-1 and DDC-4, respectively). Of the strains isolated, strain DDC-4 exhibited the highest AFB_1_ degradation rate of 76.30% ± 2.18% after 72 h of incubation at 37 °C, which was significantly higher than other strains (Figure 1).

During the physiological and biochemical tests, strain DDC-4, a Gram-positive rod bacterium, displayed the typical characteristics of *Bacillus* species (Table 1). The strain could use glucose, arabinose, xylose, mannitol, gelatin, starch, casein, citrate, and malonate and reduce nitrate and grow at pH 5.7, but it could not grow in the presence of lysozymes. Moreover, it exhibited certain temperature adaptability (30 °C–50 °C) and salt resistance (up to 10% (*w*/*v*) NaCl). Additionally, strain DDC-4 exhibited catalase activity but no phenylalanine dehydrolase or tryptophanase activity. Meanwhile, an approximately 1500 bp product was amplified from the genomic DNA of DDC-4, and a neighbor-joining tree was constructed based on the results of the 16S rRNA gene sequence analysis (Figure 2). Compared with the outgroup *Metabacillus galliciensis*, *Bacillus* species were grouped together in a single cluster. Strain DDC-4 and *B. halotolerans* ATCC 25096^T^ were clustered into the same clade with 100% sequence similarity. Thus, strain DDC-4 was recognized as *B. halotolerans*. To the best of our knowledge, this study is the first to report *B. halotolerans* as an AFB_1_-degrading strain.

### 2.2. AFB_1_ Degradation by the Active Component of Strain DDC-4 and Its Characteristics

We here investigated whether the fermentation broth, cell-free supernatant, cell suspension, and cell lysate can cause AFB_1_ degradation (Figure 3A). Overall, the cell-free supernatant removed 55.04% ± 2.60% of AFB_1_ after 72 h incubation, whereas the cell suspension and cell lysate were almost unable to remove AFB_1_, with their degradation rates being −1.88% ± 8.46% and 4.44% ± 0.52%, respectively, dramatically lower than that of the cell-free supernatant. This indicated that AFB_1_ removal by strain DDC-4 predominantly depended on degradation rather than on absorption. The cell-free supernatant was the main active component of strain DDC-4 during degradation. These findings are consistent with those of *Bacillus* species (e.g., *B. licheniformis* CFR1, *B. subtilis* UTBSP1, *B. velezensis* DY3108, and *B. amyloliquefaciens* WF2020) [7,29,30,31]. Although the cell-free supernatant had a major role in degradation, the degradation rate with the supernatant was significantly lower than that with the fermentation broth. Therefore, AFB_1_ could be speculated to exert an induction effect on the degradation activity of strain DDC-4. In other words, the expression level of genes encoding degradation-associated extracellular metabolites might be augmented under AFB_1_ stress to reduce AFB_1_-induced damage. To verify this hypothesis, we evaluated the induction effect of AFB_1_. The degradation rate of the cell-free supernatant increased to 68.08% ± 4.11% after induction. This rate was significantly higher than that of the noninduction group (50.49% ± 8.26%) and was almost the same as that of the fermentation broth (Figure 3B).

The degradation rate of the cell-free supernatant dramatically decreased to 10.60% ± 8.61% and 39.37% ± 1.18% after SDS and proteinase K pretreatments, respectively (Figure 4A). This might be because the structure of the protein in the supernatant was destroyed. By contrast, the degradation rate exhibited no decrease but increased slightly after heat treatment. The degradation rate increased more significantly when the heat treatment was prolonged. We further investigated the effect of incubation temperature on degradation activity. Similarly, the degradation rate significantly increased at 30 °C–60 °C, from 18.51% ± 1.34% to 79.24% ± 3.67%, and remained stable (approximately 80%) at 60 °C–90 °C (Figure 4B). According to these results, thermostable proteins or perhaps enzymes with a broad temperature adaptability present in the supernatant were involved in AFB_1_ degradation, and their activities were activated by heat treatment. Similar results have been observed in some AFB_1_-degrading strains, such as *B. velezensis* DY3108 [30], *B. shackletonii* L7 [32], and *P. aeruginosa* N17-1 [33], which have made them more advantageous in industrial applications.

Considering the enzyme activity loss during freeze–drying, although the cell-free supernatant was pH sensitive, it could still degrade AFB_1_ within 5–10 pH (Figure 4C). The maximum degradation rate of 45.11% ± 1.99% was observed at pH 7, while decreased significantly as the pH increased or decreased because of the impaired activity of the enzymes in the supernatant. Compared to acid, the supernatant had a stronger tolerance to alkalis. The degradation rate decreased to 30.63% ± 2.75% at pH 8 and decreased to 22.07% ± 4.22% at pH 6. The degradation rates at pH 5, pH6, pH 9 and pH 10 were not significantly different. Moreover, the activity of the cell-free supernatant was almost lost at pH 4 and pH 11, with their degradation rates being 1.63% ± 0.09% and 6.33% ± 1.92%, respectively, which were significantly lower than others.

Additionally, the effects of metal ions on degradation by the cell-free supernatant were evaluated (Figure 4D). Cu^2+^ dramatically enhanced the degradation rate to 95.88% ± 1.51%, whereas Zn^2+^ and Fe^3+^ exerted no significant effect. However, Li^+^, Ni^+^, Mg^2+^, and Ca^2+^ inhibited the degradation activity to a certain extent. Furthermore, the influence of the copper concentration revealed that the degradation rate increased sharply within the range of 0–10 mM Cu^2+^ and decreased slightly afterward (Figure 4E). Thus, Cu^2+^ might act as an activator or membrane stabilizer or an electron transfer medium for enzymes to stimulate AFB_1_ degradation activity, comparatively similar to the results obtained in other AFB_1_-degrading strains, including *B. amyloliquefaciens* WF2020 [31], *B. velezensis* DY3108 [30], and *B. shackletonii* L7 [32]. When 10 mM Cu^2+^ was added, 41.56% ± 2.02% AFB_1_ was degraded in the initial 6 h and 95.45% ± 1.81% AFB_1_ was degraded after 48 h incubation (Figure 4F). This indicated that the supernatant caused relatively rapid degradation. Moreover, the supernatant decreased the AFB_1_ content in the moldy maize powder from 6.39 ± 0.43 μg/kg to 2.96 ± 0.92 μg/kg (the degradation rate was 53.77% ± 14.42%, Table 2), which demonstrates that the cell-free supernatant of strain DDC-4 can be a potential tool for handling moldy grains.

Overall, the active components of strain DDC-4 were thermostable extracellular proteins. AFB_1_ induced the expression of genes encoding these proteins, and Cu^2+^ and heat treatment increased the activity of these proteins.

### 2.3. GO Term and KEGG Pathway Enrichment Analyses

To reveal the molecular response of DDC-4 to AFB_1_, transcriptomic analysis was performed. Q20 and Q30 values for each sample were greater than 98% and 95%, respectively (Appendix A). More than 95% of the clean reads were mapped to the reference genome (Appendix A), which indicated the reliability of the RNA sequencing results. The distance between the treated and untreated samples was significant (Appendix A). In total, 165 upregulated and 284 downregulated differentially expressed genes (DEGs; Appendix A), mapped to 27 and 32 GO terms (Appendix A), respectively, were identified after AFB_1_ treatment.

The upregulated DEGs were significantly enriched in 10 GO terms, including the histidine catabolic process, the histidine catabolic process to glutamate and formamide, the histidine catabolic process to glutamate and formate, and developmental process (Figure 5A). All the top three GO terms were related to the histidine catabolic process, with all the rich factors (the ratio of the enriched DEGs to total transcripts) being 1.00. Similar results were observed in the KEGG pathway enrichment analysis. The upregulated DEGs were significantly enriched in the histidine metabolism pathway (Figure 6A). Following AFB_1_ treatment, the expression of genes encoding histidine ammonia-lyase (EC: 4.3.1.3, encoded by the gene RS11215, HutH), urocanate hydratase (EC: 4.2.1.49, encoded by the gene RS11210, HutU), imidazolonepropionase (EC: 3.5.2.7, encoded by the gene RS11205), formimidoylglutamase (EC: 3.5.3.8, encoded by the gene RS11200), and aldehyde dehydrogenase DhaS (EC: 1.2.1.3, encoded by the gene RS20930, DhaS) was significantly upregulated to varying degrees (Figure 7 and Figure 8A). Among them, gene RS20930 was the most highly expressed, and the transcripts per million (TPM) values in the samples untreated and treated with AFB_1_ were 5215 and 11,948, respectively. The expression level of gene RS11215 showed the most significant difference between untreated and treated samples with the log2 fold change being 2.11. The induced histidine metabolism-related genes promoted the conversion of histidine to glutamate, a precursor for glutathione synthesis. Glutathione possibly participates in AFB_1_ degradation by binding to AFB_1_ or intermediate products, which is consistent with the results of Qiu et al. [34]. Furthermore, the number of DEGs enriched in the developmental process was the highest, as determined through the GO term enrichment analysis (Figure 5A). DEGs in this process were predominantly related to sporulation (Appendix A), possibly because sporulation in strain DDC-4 was promoted under AFB_1_-induced stress.

The downregulated DEGs were significantly enriched in 22 GO terms, including de novo IMP biosynthesis, IMP biosynthesis, IMP metabolism, purine nucleobase biosynthesis, purine nucleoside monophosphate biosynthesis, purine ribonucleoside monophosphate biosynthesis, and purine-containing compound biosynthesis (Figure 5B). The top three GO terms were all related to the IMP metabolic process, with all the rich factors being >0.75. Nearly all genes associated with the de novo IMP biosynthesis process (including genes RS06555, RS06560, RS06565, RS06570, RS06575, RS06580, RS06585, RS06590, RS06595, RS06600, RS06605, and RS06610) were inhibited to varying degrees (Figure 8B and Table 3). Among them, the expression of gene RS06585, RS06610, RS06590, RS06595, and RS06605 were dramatically inhibited by AFB_1_, with the log2 fold change being −2.63, −2.53, −2.39, −2.36, and −2.28, respectively. IMP serves as a precursor of AMP and GMP during de novo purine nucleobase biosynthesis. Although de novo pyrimidine nucleobase biosynthesis was not significantly inhibited, the expression of genes encoding the enzyme (carbamoyl phosphate synthase, encoded by the genes RS03965 and RS03960) involved in the first-step reaction of this process was dramatically downregulated, with the log2 fold change of the expression level being approximately −5. The expression of genes encoding hypoxanthine/guanine permease (encoded by the gene RS06640, PbuG) and uracil permease (encoded by the gene RS01625, PyrP), which might transport raw materials for the salvage pathway, was also downregulated. Similarly, the top three KEGG pathways with the highest number of enriched DEGs were purine metabolism, the two-component system, and ABC transporters, respectively (Figure 6B). These results indicated that AFB_1_ significantly inhibited nucleotide synthesis in strain DDC-4.

### 2.4. Identification and Expression Analysis of Potential Degrading Genes

The reported AFB_1_-degrading enzymes were primarily oxidoreductases. Meanwhile, hydrolase may be involved in AFB_1_ degradation from the degradation product perspective. According to our results, eight genes encoding oxidoreductases and six genes encoding hydrolases were induced following AFB_1_ treatment (Table 4). Among these genes, the gene RS11000 (aldo/keto reductase-encoding gene) was the most highly expressed (Figure 8C,D), and the TPM values in the samples untreated and treated with AFB_1_ were 305 and 930, respectively. Aldo/keto reductase, short-chain dehydrogenase/reductases (SDR) family oxidoreductase, and alpha/beta hydrolase (arylesterase), encoded by the genes RS11000, RS07845, and RS04140, respectively, possibly destroyed the lactone ring within the coumarin ring of AFB_1_ to decrease its toxicity and mutagenicity [35]. Moreover, other oxidoreductases and hydrolases might be involved in AFB_1_ degradation (Table 4), but their potential action sites need to be further investigated.

Genes RS11000, RS07845 and RS04140 were selected for the qRT-PCR analysis (Figure 9). The expression trend of these genes was consistent with the RNA-seq results, which confirmed the credibility of the transcriptomic analysis results.

## 3. Discussion

### 3.1. AFB_1_-Degrading Strains and Functional Genes

Many AFB_1_-degrading strains have been identified. Of them, the *B. subtilis* group was more sought after by researchers because of its potential probiotic characteristics and antibacterial action against *Aspergillus* species [30,31]. To our best knowledge, although several strains, including *B. subtilis* UTBSP1 [7], *B. licheniformis* CFR1 [29], *B. shackletonii* L7 [32], *B. velezensis* DY3108 [30], *B. amyloliquefaciens* WF2020 [31], and *B. albus* YUN5 [1], degrade aflatoxins, this study is the first to identify *B. halotolerans* to degrade AFB_1_. Consistent with the results of most reports about the *B. subtilis* group, the extracellular proteins of strain DDC-4 were chiefly responsible for AFB_1_ degradation. When activated with 10 mM Cu^2+^, 95.45% AFB_1_ (initial concentration: 1 μg/mL) was degraded by the extracellular proteins at 48 h, which was comparable to the results obtained with *B. velezensis* DY3108 (initial concentration: 0.5 μg/mL, >90%, 24 h) [30], *B. amyloliquefaciens* WF2020 (initial concentration: 2 μg/mL, ~100%, 48 h) [31], *B. licheniformis* CFR1 (initial concentration: 0.5 μg/mL, >90%, 24 h) [29], and *B. subtilis* UTBSP1 (initial concentration: 2.5 μg/mL, 78.39%, 72 h) [7] and higher than those obtained with *B. shackletonii* L7 [32] and *B. subtilis* JSW-1 [36]. Additionally, different from the heat-labile proteins of *B. licheniformis* CFR1 [29] and *B. subtilis* UTBSP1 [7], the active extracellular proteins of strain DDC-4 were thermostable. Furthermore, the degradation rate remained at approximately 80% at 90 °C, which was higher than those of *B. amyloliquefaciens* WF2020 [31] and *B. shackletonii* L7 [32], but slightly lower than that of *B. velezensis* DY3108 [30]. This facilitated the proteins in maintaining catalytic stability in a harsh industrial environment. The active extracellular proteins could remove 53.77% AFB_1_ from the moldy maize powder and is thus a promising agent for handling AFB_1_-contaminated food in the industry.

Although few proteins with a degradation ability have been isolated from the *B. subtilis* group, the response mechanism of this group to aflatoxins has not been completely reported. A 22-kDa heat-stable unidentified extracellular protein was purified from the cell-free supernatant of *B. shackletonii* L7 [32]. CotA laccase from *B. licheniformis* ANSB82 could transform AFB_1_ to aflatoxin Q_1_ and epi-aflatoxin Q_1_ [20]. Bacilysin biosynthesis oxidoreductase (BacC) from *B. subtilis* UTB1 was involved in AFB_1_ degradation by reducing the α,β-unsaturated ester between the lactone rings of AFB_1_ [37]. However, mass spectrometry of degradation products revealed that the difuran and lactone rings of AFB_1_ were all destroyed. Six and eight major degraded products were identified in the reaction mixture of AFB_1_ coincubated with *B. albus* YUN5 [1] and *B. subtilis* [14], respectively. Four major degraded products were detected in the *B.* sp. H16v8 and *B.* sp. HGD9229 cocultures [38]. This suggests that in addition to oxidoreductase, other types of enzymes, particularly esterase, are involved in AFB_1_ degradation. According to Pereyra et al., N-acyl-homoserine lactonase might contribute to AFB_1_ degradation [35]. However, not all AFB_1_-degrading strains of the *B. subtilis* group could produce this enzyme. In the present study, the transcriptomic analysis was performed to identify the previously neglected gene-encoding alpha/beta hydrolase (arylesterase) as the candidate gene for AFB_1_ degradation. This study provides a novel insight about AFB_1_-degrading enzymes. Alpha/beta hydrolase is a class of enzymes having similar structures and diverse functions, including esterase, lipase, proteases, and other hydrolytic enzymes [39]. Among the enzymes, arylesterase possibly targets the ester bond of AFB_1_ and thus cleaves its lactone ring to reduce its toxicity and mutagenicity.

The gene RS11000 encodes for aldo/keto reductase, which might destroy the lactone ring in AFB_1_ by reducing the keto group to the OH group. The gene RS07845 encodes for the SDR family oxidoreductase that has a broad substrate specificity. After cloning the *CgSDR* gene from *Candida guilliermondii*, Xing et al. found that recombinase transformed patulin into non-toxic E-ascladiol [40]. Thus, the SDR family oxidoreductase in this study was speculated to cleave the lactone ring in AFB_1_ following the reduction reaction catalyzed by aldo/keto reductase. Similar to the results of Xu et al., Cu^2+^ possibly serves as an electron transfer medium in redox reactions that boosts degradation activity [32]. Furthermore, all the aforementioned proteins of strain DDC-4 belonged to the general stress protein, which could confer advantages to bacteria under stress, such as salt, osmosis, oxidative damage, and freezing [41]. In this study, the expression of genes RS11000 and RS07845 was significantly upregulated under AFB_1_ stress, which might allow the strain to survive in the presence of the toxicological effects of AFB_1_ because these genes are associated with AFB_1_ degradation.

Glutathione exerted its detoxification effect on AFB_1_ by binding to it or its intermediate products, and this was first observed in mammals. In a reaction mixture of AFB_1_ coincubated with *A. niger* FS10, Qiu et al. analyzed AFB_1_ degradation products through triple quadrupole-linear ion trap-mass spectrometry (Q-Trap-MS) coupled with LightSight™ software (Version 2.2.1) [34]. They found that glutathione formed AFB_2_-GOH (C_27_H_31_N_3_O_13_) with AFB_1_ to modify the toxicity site of AFB_1_. As mentioned above, glutathione might participate in the AFB_1_ degradation of strain DDC-4. As glutamate is a precursor for glutathione synthesis, the conversion of histidine to glutamate was promoted in the AFB_1_-treated samples (Figure 10).

Several potential mycotoxins degradation genes were also selected by transcriptomic analysis due to their upregulated expression in the present of mycotoxins, such as short-chain aryl-alcohol dehydrogenase for patulin degradation [42] and carboxypeptidase A4 for ochratoxin A degradation [26]. However, the specific function of these genes in mycotoxins degradation still needs to be validated by heterologous expression. The degradation mechanism will be revealed by analysis of degradation products of recombinant protein expressed in host strain. The encoding gene of an acyl coenzyme A thioester hydrolase in *B. amyloliquefaciens* H6 was selected from upregulated genes under zearalenone stress by transcriptomic analysis [24]. The recombinant protein was expressed in *Escherichia coli*. The purified recombinant protein could convert zearalenone to the less toxic metabolites by cleaving the lactone bond and breaking down its macrolide ring [24]. More experiments will be carried out in our future study.

### 3.2. Toxicological Effect of AFB_1_ on Nucleic Acid Synthesis

On measuring the incorporation of me-[^3^H] thymidine and 6-[^14^C] orotic acid into DNA and RNA, respectively, Butler and Neal found that AFB_1_ inhibited nucleic acid synthesis [43]. Numerous subsequent studies have supported this viewpoint. However, the underlying molecular mechanism remains unclear. We here conjectured that AFB_1_ inhibited nucleic acid synthesis in strain DDC-4 through two hypothetical pathways (Figure 10). First, AFB_1_ inhibited nucleotide synthesis in strain DDC-4. As shown previously, AFB_1_ significantly inhibited de novo nucleotide biosynthesis by suppressing the expression level of genes encoding enzymes involved in this process. Moreover, the salvage pathway might be inhibited by reducing the transportation of raw materials. Second, AFB_1_ inhibited DNA replication in strain DDC-4. The process of DNA replication is divided into three stages: initiation, extension, and termination. At the beginning of extension, short RNA fragments (called primers), which are synthesized by primase, are acted as a starting point for DNA polymerase III. After termination, the primers are removed by ribonuclease H and DNA polymerase I (Appendix A). In this study, the expression level of genes encoding ribonuclease H, including ribonuclease HI (encoded by the gene RS20480) and ribonuclease HIII (encoded by the gene RS17285), were inhibited by AFB_1_; thereby, DNA replication was inhibited.

AFB_1_ was bioactivated by cytochrome P450 to generate the intermediate AFB_1_-8,9-epoxide [44]. This intermediate product was then attacked by N^7^ of guanine to form *trans*-8,9-dihydro-8-(N^7^-guanyl)-9-hydroxyaflatoxin B_1_ (AFB_1_-N^7^-Gua). This was considered as the main AFB_1_–DNA adduct causing mutations (Figure 10). Nucleotide excision repair (NER) is a pivotal player in removing AFB_1_–DNA damage in both bacterial and mammalian systems [44]. In prokaryotes, the Uvr system is involved in NER. In the present study, the expression level of the UvrD gene was upregulated after AFB_1_ treatment, whereas that of the UvrABC gene remained almost unchanged.

Altogether, AFB_1_ inhibited the synthesis of nucleotides, including purines and pyrimidines, and DNA replication (Figure 10). Cytochrome P450-mediated mutations may increase following AFB_1_ treatment. In the case of the resistance and adaptation to AFB_1_, the expression of genes encoding the potential AFB_1_-degrading enzyme was upregulated, and sporulation in strain DDC-4 was promoted.

Although several potential AFB_1_-degrading enzymes were selected, further validation of their function is needed. In the future, we will obtain the aforementioned enzymes through heterologous expression and purification. Degradation activity of the enzymes will be verified, the structures of the degradation products will be determined, and the safety of degradation products will be evaluated.

## 4. Conclusions

In this study, a novel AFB_1_-degrading strain was isolated and identified as *B. halotolerans* DDC-4 (belonging to the *B. subtilis* group). The active components of this strain were thermostable extracellular proteins or enzymes with a wide temperature adaptability. More than 90% AFB_1_ was degraded by the proteins or enzymes when Cu^2+^ was added. Thus, after adequate purification, these enzymes or proteins could serve as promising agents for AFB_1_ biodegradation in the food industry. To our best knowledge, this study is the first to explore response mechanisms of the *B. subtilis* group to AFB_1_ through transcriptomic analysis. Inhibition of nucleic acid synthesis was the primary toxicological effect of AFB_1_ on strain DDC-4. To survive under this stress, sporulation was promoted in the bacteria and the expression of genes encoding these degradation-related enzymes were induced. The genes encoding alpha/beta hydrolase (arylesterase), aldo/keto reductase, and SDR family oxidoreductase were selected as candidate genes for AFB_1_ degradation. Our study will be helpful to reveal the degradation mechanism of AFB_1_ and provide more options for handling AFB_1_-contaminated food.

## 5. Materials and Methods

### 5.1. Isolation of AFB_1_-Degrading Strains

First, 10 g moldy maize and 10 g moldy rice were separately diluted in 90 mL sterile distilled water and incubated in water bath shaker (Guangdong Foheng Instrument Co., Ltd., Dongguan, China) at 37 °C with continuous shaking (150 rpm) for 72 h. The samples were serially diluted to 10^−7^ with sterile distilled water. Aliquots (150 µL) of each dilution or strains stored in our lab (isolated from Chinese traditional fermented foods) were spread on plates containing modified Hormisch medium (HM: 0.1% coumarin, 0.05% KNO_3_, 0.05% (NH_4_)_2_SO_4_, 0.025% KH_2_PO_4_, 0.025% MgSO_4_·7H_2_O, 0.0005% CaCl_2_, 0.0003% FeCl_3_·6H_2_O, 2% agar) [27]. Each plate was cultured at 37 °C for 7 days. Visible single colonies were isolated and transferred to fresh HM plates. The aforementioned process was repeated 3–5 times until pure isolates were obtained.

To test the AFB_1_ degradation activity, each pure isolate was inoculated in Luria-Bertani (LB) medium, cultivated overnight at 37 °C with continuous shaking (150 rpm), and diluted to an optical density at 600 nm (OD_600_) of 0.4. The medium was modified to maintain neutrality during fermentation. Then, 500 μL of each dilution was added to the modified LB medium (1% peptone, 1% NaCl, 0.5% yeast extract, 0.1% KH_2_PO_4_). Fermentation was carried out at 37 °C for 48 h by shaking. Then, 960 μL of the fermentation broth was co-incubated with 40 μL of 25 μg/mL AFB_1_ (Yuanye Bio-Technology Co., Ltd., Shanghai, China) in the dark at 37 °C for 72 h with shaking (150 rpm). Sterile modified LB medium containing AFB_1_ was used as the control. The supernatant was recovered through centrifugation at 4500 rpm for 10 min at room temperature. Subsequently, 650 μL of the supernatant was mixed with 350 μL methanol, and residual AFB_1_ was analyzed using the ELISA kit (Youlong Biotech Co., Ltd., Shanghai, China). According to kit instruction, the cross-reactivity ration with similar toxin AFB_2_, AFG_1_, and AFG_2_ was 13%, 1.9%, and 5.7%, respectively, indicating the kit could specifically detect AFB_1_. The AFB_1_ degradation rate was calculated as follows:(1)$$Y=\left(X_{1}-X_{2}\right)/X_{1}\times 100\%$$
where *X*_1_ is the residual AFB_1_ in the control, *X*_2_ is the residual AFB_1_ in the sample, and *Y* is the AFB_1_ degradation rate (%).

### 5.2. Identification of Strain DDC-4

Strain DDC-4 was identified through physiological and biochemical tests and 16S rRNA gene sequencing. The physiological and biochemical tests were conducted using the specified reagents (Haibo Biotechnology Co., Ltd., Qingdao, China). Meanwhile, genomic DNA was extracted using the E.Z.N.A Bacterial DNA Kit (Omega Bio-tek. Inc., Norcross, GA, USA). The 16S rRNA-coding gene was amplified through PCR by using the universal primer pair 16S-F and 16S-R (Appendix A) [45], sequenced by General Biosystems Co., Ltd. (Chuzhou, China), aligned with sequences found on the EzBioCloud server [46], and deposited in the NCBI GenBank with accession number OQ306542. A phylogenetic tree was constructed with MEGA software (version 6.0) using the neighbor-joining method [47].

### 5.3. AFB_1_ Degradation by the Cell-Free Supernatant, Cell Suspension, and Cell Lysate

Strain DDC-4 was fermented as mentioned above. After fermentation for 48 h, 2 mL fermentation broth was centrifuged at 4500 rpm for 10 min at room temperature to separate the cell-free supernatant and cells. The cells were washed with 2 mL phosphate buffer saline (PBS: 137 mM NaCl, 2.7 mM KCl, 4.5 mM Na_2_HPO_4_, 1.4 mM KH_2_PO_4_) twice and resuspended in 2 mL PBS. Then, the solution was divided into two fractions. One fraction was processed without any treatment (namely, cell suspension). The other fraction was disintegrated through ultrasonication (Sonics, Newtown, Connecticut, USA, 50% of maximum amplitude, subjected to ultrasound for 3 min with a 5 s interval between two 3 s processing) in the ice bath and centrifuged at 10,000 rpm for 2 min at 4 °C to obtain the supernatant (namely, cell lysate). The obtained cell-free supernatant and cell lysate were separately filtered through a 0.45 µm filter. Then, 960 μL each of the fermentation broth, cell-free supernatant, cell suspension, and cell lysate was separately coincubated with 40 μL of 25 μg/mL AFB_1_. The sterile modified LB medium or PBS containing AFB_1_ served as the control. Residual AFB_1_ in each sample was determined as described previously. The AFB_1_ degradation rate was calculated using the aforementioned formula.

### 5.4. Induction Effect of Degradation by AFB_1_

Strain DDC-4 was fermented as mentioned above. The fermentation broth was divided into four fractions, namely fraction A, fraction B, fraction C, and fraction D. To investigate the induction effect of AFB_1_ on degradation by comparing the degradation rate of the cell-free supernatant induced by AFB_1_, the cell-free supernatant uninduced by AFB_1_, and the fermentation broth; the AFB_1_ addition concentration and incubation time were the same as those used while determining the degradation rates of the fermentation broth.

Fraction A (induced by AFB_1_): 960 μL of the fermentation broth was treated with 40 μL of 25 μg/mL AFB_1_ in the dark at 37 °C for 72 h with shaking (150 rpm). The supernatant was collected through centrifugation at 4500 rpm for 10 min at room temperature, filtered through a 0.45 µm filter, and coincubated with 40 μL of 25 μg/mL AFB_1_ in the dark at 37 °C for 72 h with shaking (150 rpm).

Fraction B (without induction): 960 μL of the fermentation broth was treated with 40 μL sterile distilled water in the dark at 37 °C for 72 h with shaking (150 rpm). The supernatant was treated in the same manner as fraction A.

Fraction C: 960 μL of the fermentation broth was treated with 40 μL of 25 μg/mL AFB_1_ in the dark at 37 °C for 72 h with shaking (150 rpm).

Fraction D: 960 μL of the sterile modified LB medium was coincubated with 40 μL of 25 μg/mL AFB_1_ in the dark at 37 °C for 72 h with shaking (150 rpm).

Residual AFB_1_ in each fraction was determined as described above. The AFB_1_ degradation rate was calculated as follows:(2)$$Y_{i}=\left(C+D-A\right)/\left(C+D\right)\times 100\%$$
(3)$$Y_{u}=\left(D-B\right)/D\times 100\%$$
where *A* is the residual AFB_1_ in Fraction A, *B* is the residual AFB_1_ in Fraction B, *C* is the residual AFB_1_ in Fraction C, *D* is the residual AFB_1_ in Fraction D, *Y_i_* is the AFB_1_ degradation rate of the induction group (%), and *Y_u_* is the AFB_1_ degradation rate of the noninduction group (%).

### 5.5. Effects of Heat, SDS, and Proteinase K Treatments on AFB_1_ Degradation by the Cell-Free Supernatant

The cell-free supernatant was prepared as mentioned above. To investigate the effects of heat, SDS, and proteinase K treatments, the cell-free supernatant was treated with boiling water for 10 and 30 min, 1% SDS in the dark for 24 h, and 1 mg/mL proteinase K in the dark for 24 h, respectively. The degradation experiment was conducted as mentioned above. The sterile modified LB medium containing AFB_1_ was used as the control.

### 5.6. Effects of Incubation Conditions on AFB_1_ Degradation by the Cell-Free Supernatant

The cell-free supernatant was prepared as mentioned above. To demonstrate the effects of temperature, the supernatant containing AFB_1_ was incubated at different temperatures (30 °C, 40 °C, 50 °C, 60 °C, 70 °C, 80 °C, and 90 °C) without shaking in the dark for 72 h. In the pH test, the supernatant was freeze-dried, redissolved in an equal volume of different buffers (citrate buffer (pH 4 and 5), phosphate buffer (pH 6, 7 and 8), and sodium carbonate/sodium bicarbonate buffer (pH 9, 10, and 11)), and coincubated with AFB_1_ in the dark at 37 °C for 72 h with shaking. Regarding metal ions, the supernatant was added to 10 mM each of Li^+^ (LiCl), Ni^2+^ (NiSO_4_), Cu^2+^ (CuSO_4_), Mg^2+^ (MgCl_2_), Ca^2+^ (CaCl_2_), Zn^2+^ (ZnSO_4_), Mn^2+^ (MnCl_2_), and Fe^3+^ (FeCl_3_) and coincubated with AFB_1_ in the dark at 37 °C for 72 h with shaking. The influence of the copper concentration (1, 5, 10, and 15 mM) and that of incubation times (6, 12, 18, 24 h, 36, 48, and 72 h) with 10 mM Cu^2+^ on AFB_1_ degradation were also determined. The residual AFB_1_ in each sample was determined as mentioned above, and the sterile modified LB medium substituted the supernatant in the control.

### 5.7. Application of the Cell-Free Supernatant to Remove AFB_1_ from the Moldy Maize Powder

After the moldy maize powder was sterilized, 5 g of the powder was mixed with 10 mL cell-free supernatant of strain DDC-4 and 10 mM Cu^2+^ and incubated for 48 h. The AFB_1_ content was analyzed using the ELISA kit.

### 5.8. RNA Extraction and Sequencing

First, 960 μL of the fermentation broth was treated separately with 40 μL of 25 μg/mL AFB_1_ and 40 μL sterile distilled water in the dark at 37 °C for 72 h with shaking. Three independent biological replicates were used for each treatment. The cells were obtained through centrifugation at 5000× *g* for 10 min. Total RNA was extracted using the Total RNA Extractor Kit (Sangon Biotech Co., Ltd., Shanghai, China). RNA quality and integrity was detected through 1% agarose gel electrophoresis, and the RNA concentration was determined using the NanoDrop (Thermo Fisher Scientific, Inc., Waltham, MA, USA). The concentration and quality of RNA met the requirements for libraries construction (Appendix A). rRNAs were removed using the Ribo-off rRNA Depletion kit (Vazyme Biotech Co., Ltd., Nanjing, China). cDNA libraries were constructed using the VAHTS™ Stranded mRNA-seq Library Prep Kit for Illumina^®^ (Vazyme Biotech Co., Ltd., China). Library quality was examined through 8% polyacrylamide gel electrophoresis. The libraries were sequenced on the DNBseq-T7 (BGI Genomics Co., Ltd., Shenzhen, China) platform (Sangon Biotech Co., Ltd., China) to obtain raw reads. The clean reads were acquired using the Trimmomatic program (version 0.36) for data processing. After the reads were evaluated for quality, the clean reads were mapped to the reference genome of *B. halotolerans* ZB201702 from the NCBI database (https://www.ncbi.nlm.nih.gov/assembly/GCF_004006435.1/?shouldredirect=false, accessed on 9 January 2019) using the Bowtie2 program (version 2.3.2). The transcriptome sequencing data were stored in the Sequence Read Archive (https://dataview.ncbi.nlm.nih.gov/object/PRJNA917813?reviewer=jk3r3v6iq68bb7bch1o9esdkn9, created on 4 January 2023).

### 5.9. GO Term and KEGG Pathway Enrichment Analyses

Heatmaps were constructed using the gplots package in R to present the distance between the samples. Transcripts per million (TPM) values were calculated using the featureCounts program (version 1.6.0) to reflect the gene expression level. Differentially expressed genes (DEGs) between the samples untreated and treated with AFB_1_ were selected using the DESeq2 (version 1.12.4) package in R while considering |log2 fold change| > 1 and *q* value < 0.05 as the filtering criteria. Functions of DEGs were annotated by referring to bioinformatics databases, including the Nonredundant Protein, Gene Ontology (GO), the Kyoto Encyclopedia of Genes and Genomes (KEGG), and Cluster of Orthologous Groups of Proteins databases. GO term and KEGG pathway enrichment analyses were performed using topGO (version 2.24.0) and the clusterProfiler (version 3.0.5) package in R, respectively. The significance level was determined using the *q* value (<0.05). The expression of the selected genes was presented in the heatmaps constructed using the pheatmap package in R.

### 5.10. Quantitative Real-Time PCR

To validate the RNA-seq results, genes RS11000, RS07845, and RS04140 were selected as target genes and examined through quantitative real-time PCR (qRT-PCR). Primers were designed using Premier 6 (Appendix A). Total RNA was transcribed into cDNA using the PrimeScript™ 1st Strand cDNA Synthesis Kit (Takara, Dalian, China). qRT-PCR was performed using SYBR^®^ Premix Ex TaqTM (Takara, Dalian, China) on the 7500 Real-Time PCR System (ABI, Foster City, CA, USA). Relative expression levels of the target genes were normalized by the expression levels of the internal control gene (16S rRNA) and quantified using the ΔΔCt method. Three independent biological replicates were used.

### 5.11. Statistical Analysis

All assays were conducted in triplicate. The study results are expressed as mean ± SD and analyzed conducting Duncan’s multiple comparison test (*p* < 0.05) with SPSS software (version 22.0.0.0).

## Figures and Tables

**Figure 1 toxins-16-00256-f001:**
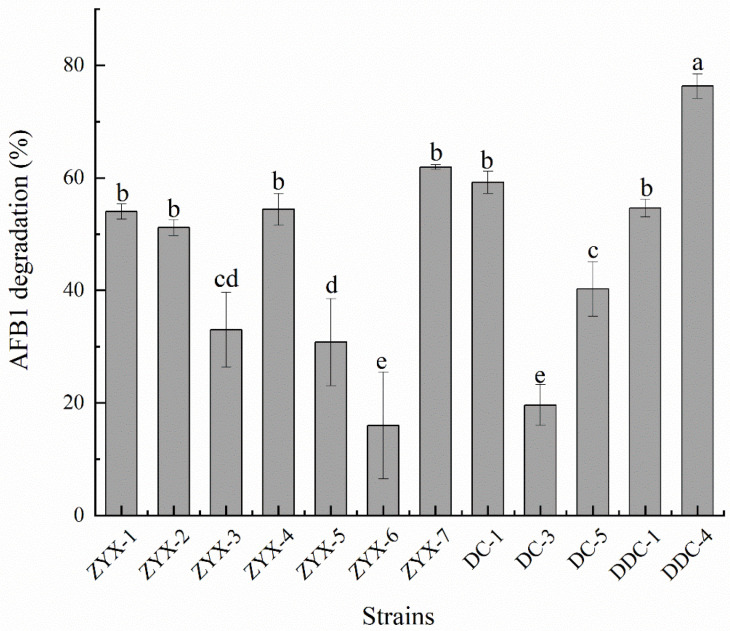
The aflatoxin B_1_ (AFB_1_) degradation rates of isolated strains co-incubated with AFB_1_ in the dark at 37 °C for 72 h with shaking. Each value is presented as the mean ± SD (*n* = 3). Different letters represent significant differences between species (*p* < 0.05).

**Figure 2 toxins-16-00256-f002:**
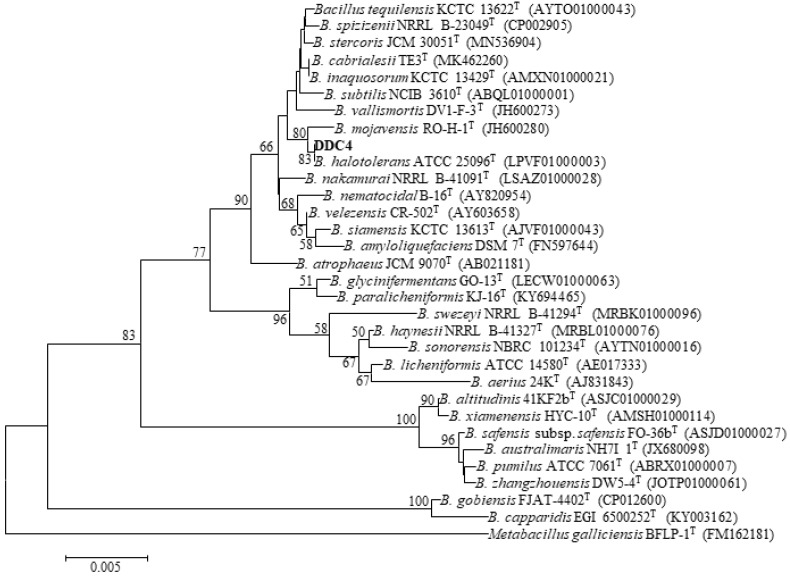
Neighbor-joining tree reconstructed using 16S rRNA gene sequences from the EzBioCloud server by MEGA software version 6.0. *Metabacillus galliciensis* was used as an outgroup. Numbers at branches indicate bootstrap values (>50%) from 1000 replicates.

**Figure 3 toxins-16-00256-f003:**
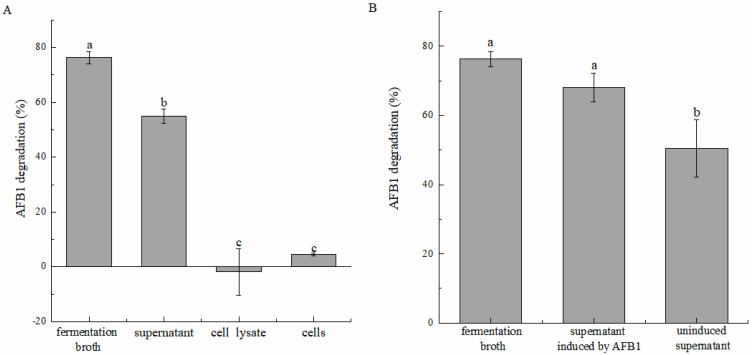
Degradation of AFB_1_ by different components of strain DDC-4 in the dark at 37 °C for 72 h (**A**), and AFB_1_-induced enhancement effect (**B**). Each value is presented as the mean ± SD (*n* = 3). Different letters represent significant differences between species (*p* < 0.05).

**Figure 4 toxins-16-00256-f004:**
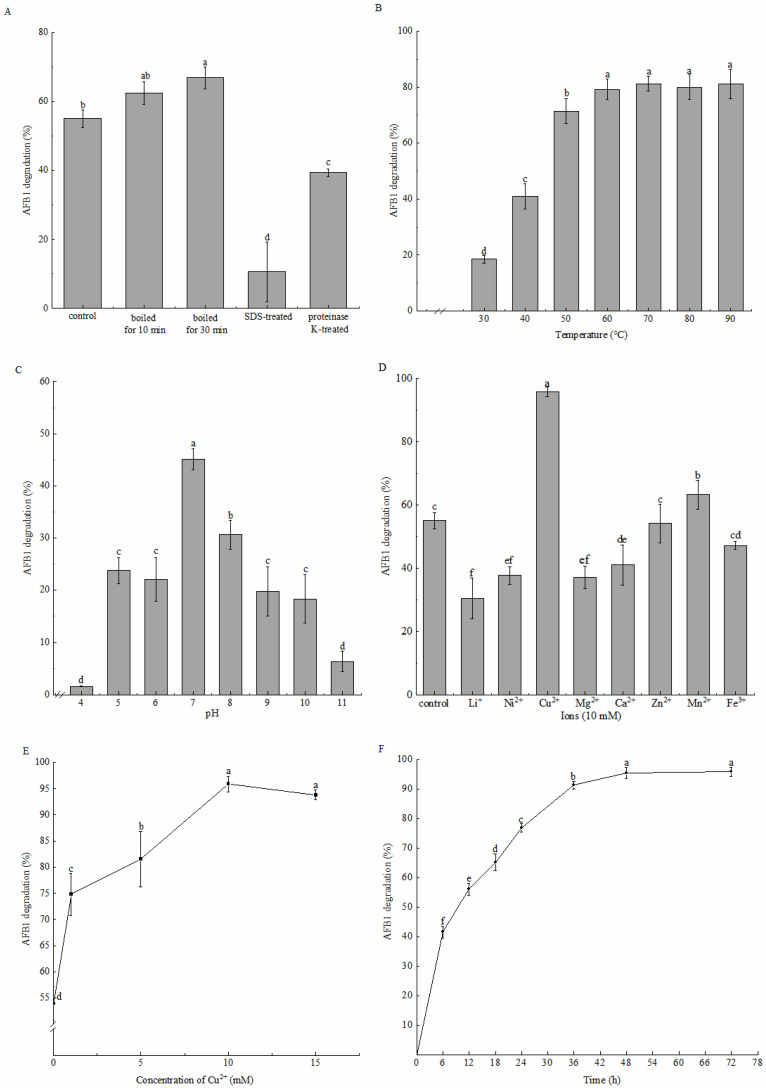
The influences of different factors on AFB_1_ degradation by the cell-free supernatant of strain DDC-4. The influence of heat, SDS and proteinase K treatments (**A**), temperature (**B**), pH (**C**), metal ions (**D**), copper concentration (**E**), and incubation time (**F**) are shown. Each value is presented as the mean ± SD (*n* = 3). Different letters represent significant differences between species (*p* < 0.05).

**Figure 5 toxins-16-00256-f005:**
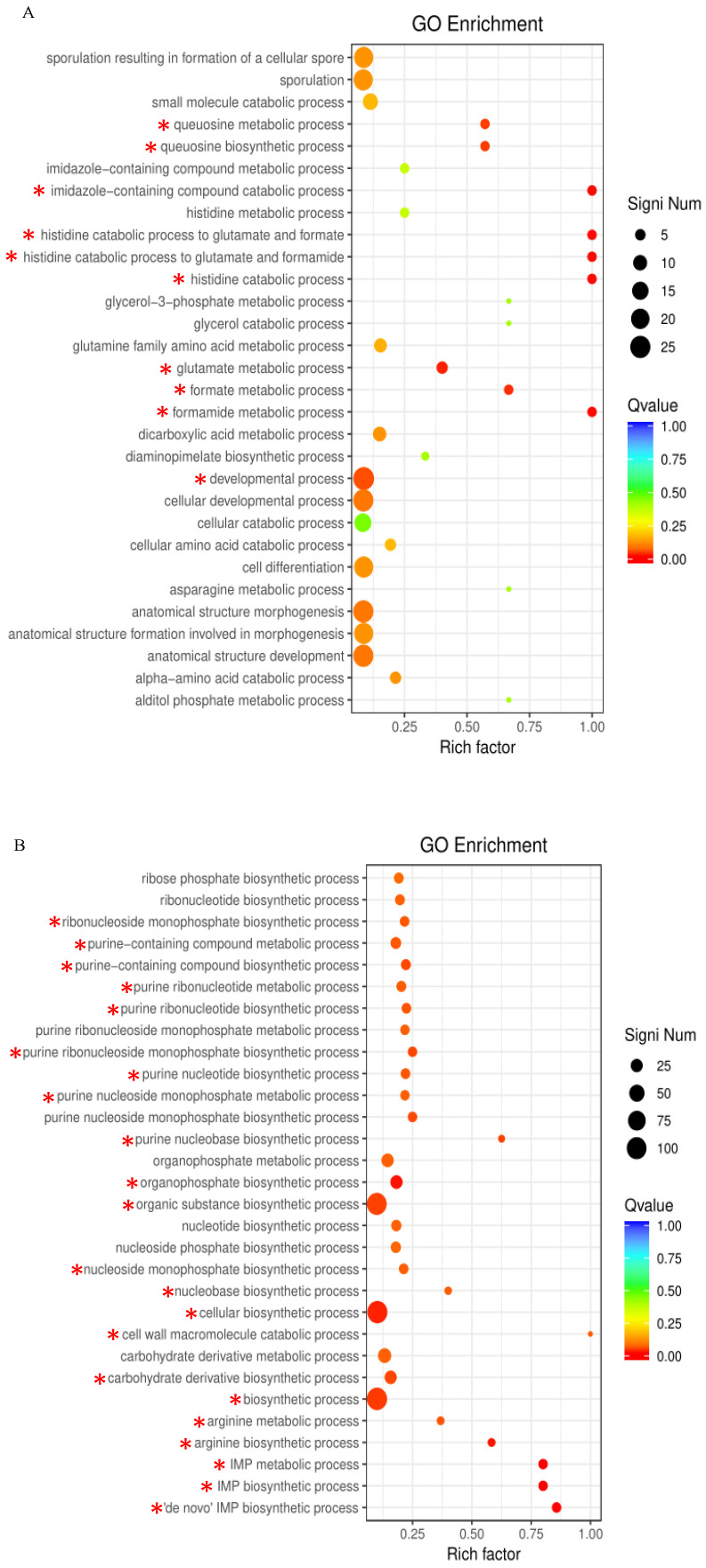
GO term enrichment analysis of upregulated (**A**) and downregulated (**B**) differentially expressed genes (DEGs). The circle size indicates the number of DEGs enriched in each pathway. The Q value indicates the significance of enrichment, increasing from blue to red. Rich factor represents the ratio of the enriched DEGs to total transcripts in this pathway. *, represents the DEGs in significantly enriched pathways.

**Figure 6 toxins-16-00256-f006:**
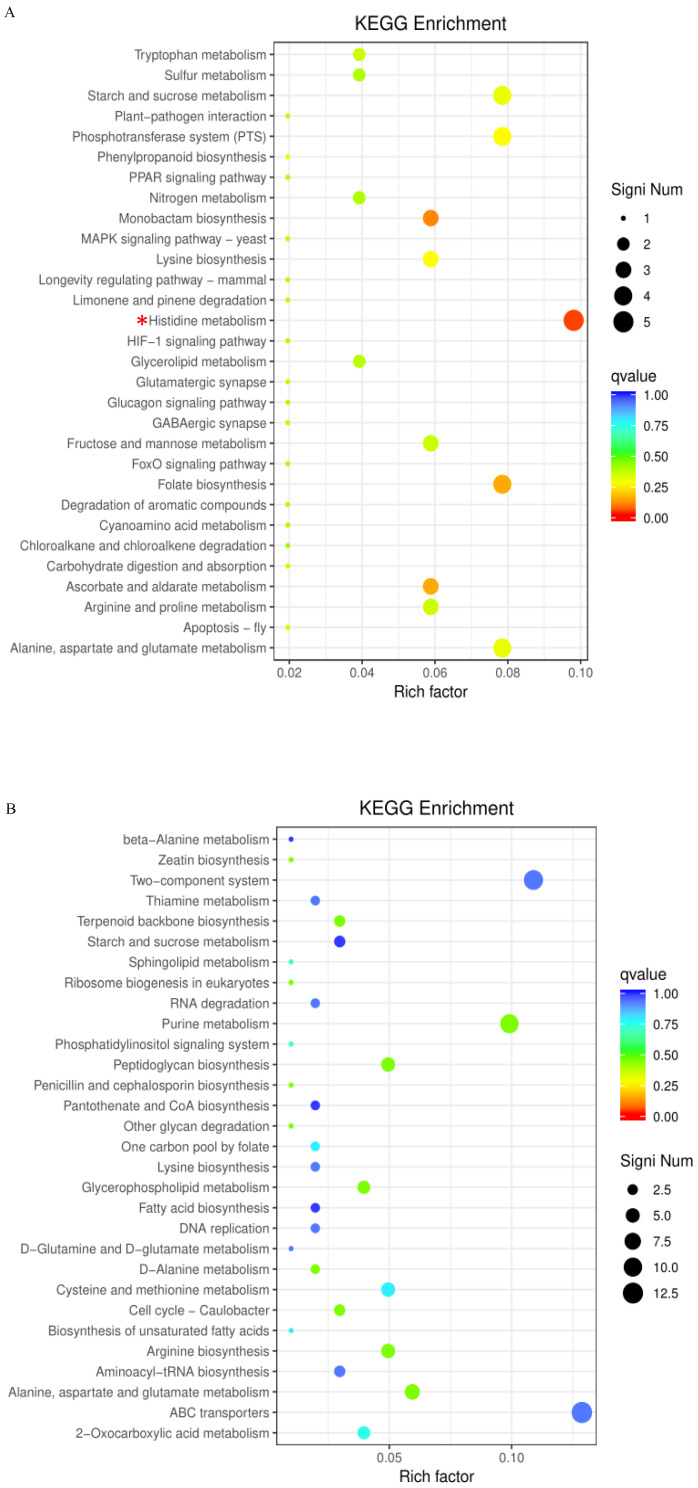
KEGG pathway enrichment analysis of upregulated (**A**) and downregulated (**B**) DEGs. The circle size indicates the number of DEGs enriched in each pathway. The Q value indicates the significance of enrichment, increasing from blue to red. Rich factor represents the ratio of the enriched DEGs to total transcripts in this pathway. *, represents the DEGs in significantly enriched pathways.

**Figure 7 toxins-16-00256-f007:**
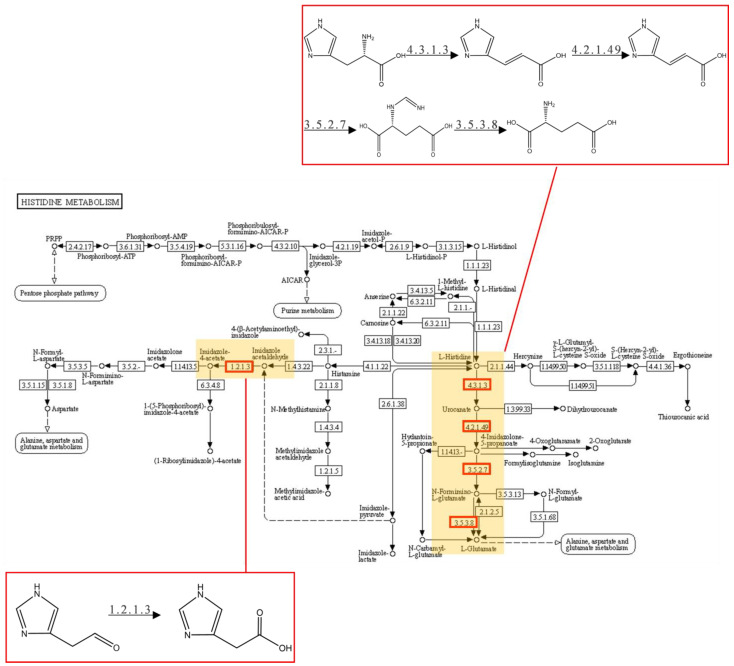
The pathway of histidine metabolism. The red rectangle indicates the enzyme-encoding gene induced by AFB_1._ The reaction substrates and products of these enzymes are also shown.

**Figure 8 toxins-16-00256-f008:**
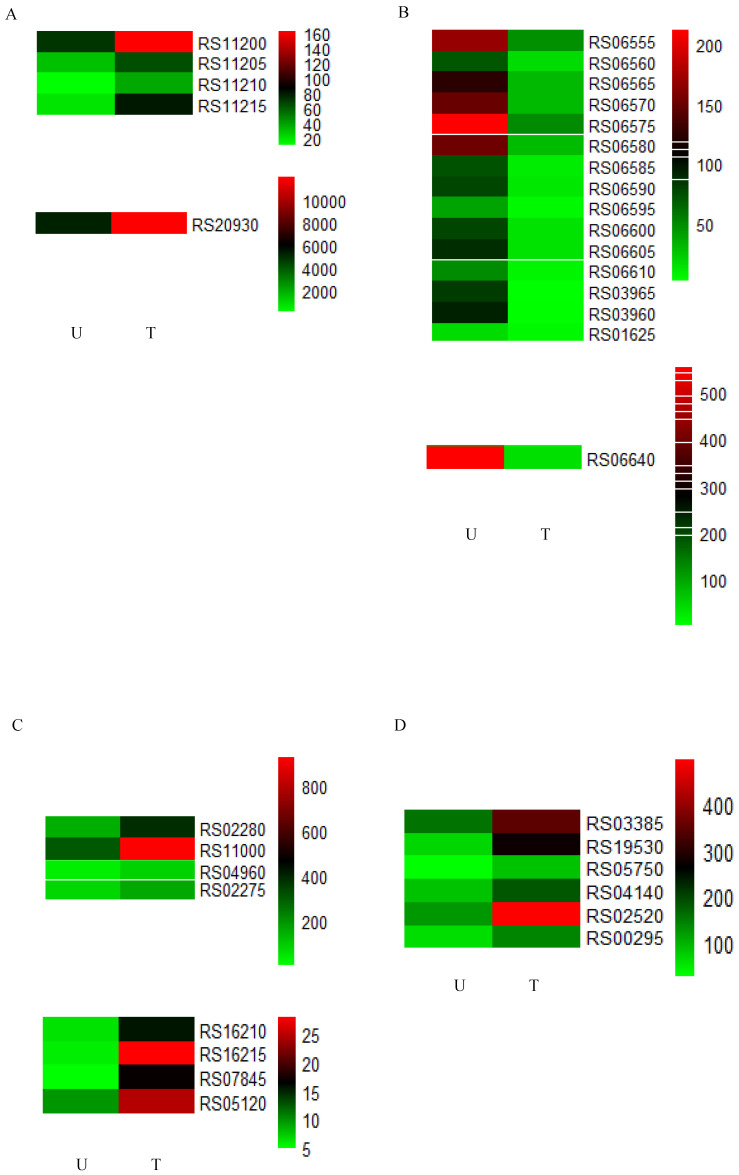
Expression patterns of DEGs in the pathway of histidine metabolism (**A**), DEGs in the pathway of ‘de novo’ IMP biosynthetic process (**B**), upregulated oxidoreductase encoding genes (**C**), and upregulated hydrolase encoding genes (**D**). Different colors represent different expression levels (increasing from green to red). U and T indicate AFB_1_-untreated and -treated samples, respectively.

**Figure 9 toxins-16-00256-f009:**
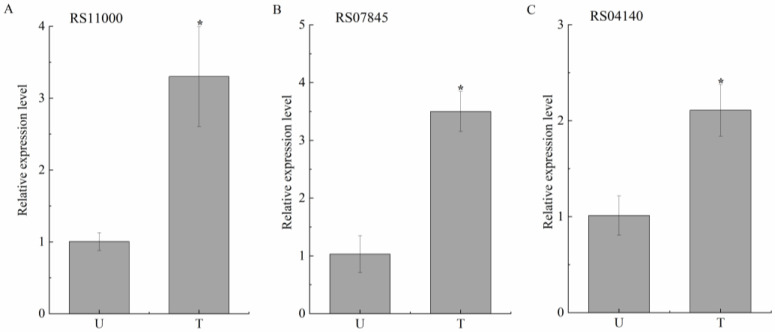
Relative expression levels of gene RS11000 (**A**), RS07845 (**B**), and RS04140 (**C**) between AFB_1_-treated and -untreated samples based on qRT-PCR analysis. U and T represent AFB_1_-treated and -untreated samples, respectively. 16S rRNA was used as an internal control. Each value is presented as the mean ± SD (*n* = 3). *, representssignificant differences between species (*p* < 0.05).

**Figure 10 toxins-16-00256-f010:**
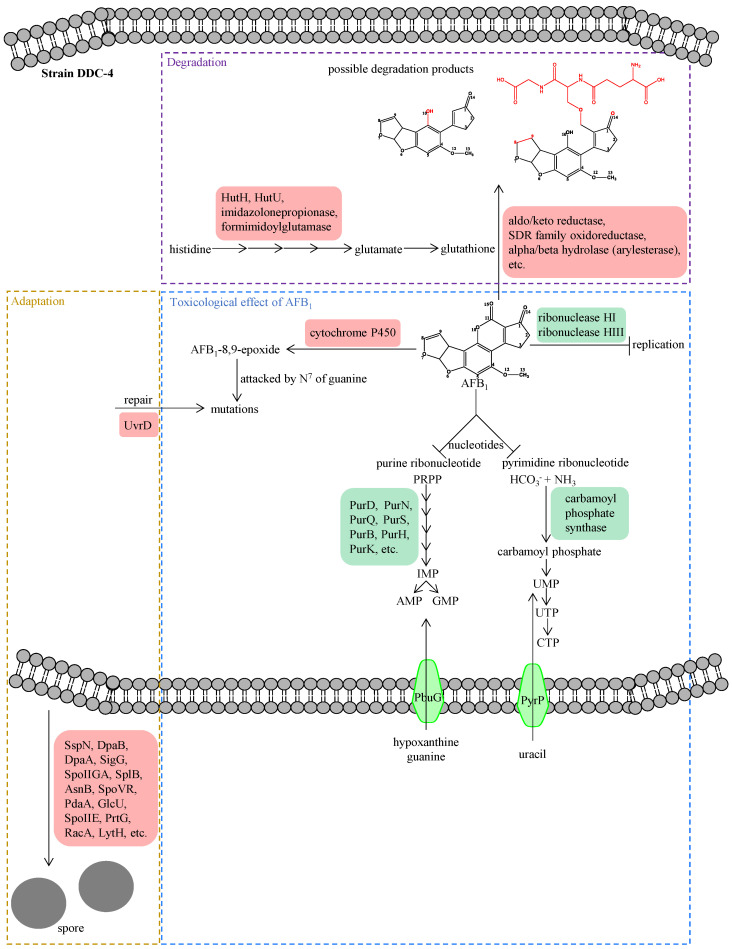
Proposed response mechanisms of strain DDC-4 to AFB_1_. Up- and downregulated encoding genes were displayed in red and green fillings, respectively.

**Table 1 toxins-16-00256-t001:** Physiological and biochemical characteristics of strain DDC-4.

Items	DDC-4
Gram stain	Gram-positive rod
Moveability	+
Voges-Proskauer	+
Oxidation of	
glucose	+
arabinose	+
xylose	+
mannitol	+
Hydrolysis of	
gelatin	+
starch	+
casein	+
Growth on	
citrate	+
lysozyme	−
5 °C	−
10 °C	−
30 °C	+
40 °C	+
50 °C	+
55 °C	−
65 °C	−
NaCl (2%)	+
NaCl (5%)	+
NaCl (7%)	+
NaCl (10%)	+
pH 5.7	+
Phenylalanine dehydrolase	−
Catalase activity	+
Nitrate reduction	+
Malonate	+
Indole	−

“+” and “−” indicates that the result is positive and negative, respectively.

**Table 2 toxins-16-00256-t002:** The content of aflatoxin B_1_ (AFB_1_) in moldy maize powder.

Sample Name	The Content of AFB_1_ (μg/kg)
Initial	6.76 ± 0.85 ^a^
Sterilization	6.39 ± 0.43 ^a^
Treatment	2.96 ± 0.92 ^b^

Initial: moldy maize powder without treatment; sterilization: moldy maize powder was sterilized in an autoclave; treatment: moldy maize powder was mixed with the cell-free supernatant of strain DDC-4 and 10 mM Cu^2+^ for 48 h after sterilization.

**Table 3 toxins-16-00256-t003:** Differentially expressed genes (DEGs) enriched in the ‘de novo’ purine nucleobase biosynthetic process.

Gene id	Gene Name	Gene Description
RS06555	*PurD*	phosphoribosylamine-glycine ligase
RS06560	*PurH*	IMP cyclohydrolase
RS06565	*PurN*	phosphoribosylglycinamide formyltransferase
RS06570	RS06570	phosphoribosylformylglycinamidine cyclo-ligase
RS06575	RS06575	amidophosphoribosyltransferase
RS06580	*PurL*	phosphoribosylformylglycinamidine synthase subunit PurL
RS06585	*PurQ*	phosphoribosylformylglycinamidine synthase subunit PurQ
RS06590	*PurS*	phosphoribosylformylglycinamidine synthase subunit PurS
RS06595	RS06595	phosphoribosylaminoimidazolesuccinocarboxamide synthase
RS06600	*PurB*	adenylosuccinate lyase
RS06605	*PurK*	5-(carboxyamino)imidazole ribonucleotide synthase
RS06610	*PurE*	5-(carboxyamino)imidazole ribonucleotide mutase

**Table 4 toxins-16-00256-t004:** Genes encoding oxidoreductase and hydrolase induced by AFB_1_ in strain DDC-4.

Gene id	Gene Name	Gene Description
RS16210	RS16210	cytochrome ubiquinol oxidase subunit II
RS16215	RS16215	cytochrome ubiquinol oxidase subunit I
RS02280	*AhpA*	biofilm-specific peroxidase AhpA
RS11000	RS11000	aldo/keto reductase
RS07845	RS07845	SDR family oxidoreductase
RS04960	RS04960	NAD(P)H-dependent oxidoreductase
RS05120	RS05120	NAD(P)H-dependent oxidoreductase
RS02275	*YkuV*	thiol-disulfide oxidoreductase YkuV
RS03385	RS03385	NUDIX hydrolase
RS19530	RS19530	alpha/beta hydrolase (haloalkane dehalogenase)
RS05750	RS05750	amidohydrolase
RS04140	RS04140	alpha/beta hydrolase (arylesterase)
RS02520	RS02520	glycoside hydrolase family 18 protein
RS00295	RS00295	poly-gamma-glutamate hydrolase family protein

Potential degrading genes are shown in red.

## Data Availability

The original contributions presented in the study are included in the article and Appendix A, further inquiries can be directed to the corresponding authors.

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
