# Peer review of "Identification of a Novel Aflatoxin B1-Degrading Strain, Bacillus halotolerans DDC-4, and Its Response Mechanisms to Aflatoxin B1"

_toxins, 2024, doi:10.3390/toxins16060256_

Round 1
Reviewer 1 Report
Comments and Suggestions for Authors
Journal Toxins (ISSN 2072-6651)
Manuscript ID toxins-3013979
Type
Article: “Identification of a novel aflatoxin B1-degrading strain and its response mechanisms to aflatoxin B1”
Section Mycotoxins
The paper is very good. The authors make a presentation supported by well-developed and explained essays. It is a topic of scientific interest, which should be accepted for publication after some minor suggestions given below:
1- The name of the Bacillus halotolerans strain should be included in the title of the paper.
2- 5. Materials and Methods (Lines 421-424)
5.1. Isolation of AFB1-degrading strains
First, 10 g moldy maize and 10 g moldy rice were separately diluted in 90 mL sterile distilled water and incubated at 37 ℃ with continuous shaking (150 rpm) for 72 h.
The equipment used, brand and country of origin should be included
3- The Author Contributions section is a requirement of MDPI Journals, please include this section in the paper.
Reviewer 2 Report
Comments and Suggestions for Authors
This manuscript investigates a novel aflatoxin B1 (AFB1)-degrading Bacillus halotolerans strain (DDC-4) and its response mechanisms. Given below is the assessment of the manuscript and suggestions for its improvement
Strengthen the introduction by providing a more focused and up-to-date review of AFB1 degradation mechanisms, particularly highlighting the novelty of using alpha/beta hydrolase.
Specificity of the ELISA kit for AFB1 detection should be mentioned.
The rationale behind using coumarin as the sole carbon source for isolation needs explanation.
Justification for the chosen concentration and incubation time for AFB1 induction is required.
Include statistical analysis of the results to demonstrate their significance.
Add controls for RNA extraction and sequencing to validate the transcriptome data.
The results are well presented; however, some parts need clarifications in their presentations.
The discussion needs to be improved.
Section 3.1: Discuss the limitations of using transcriptomic data as the sole evidence for enzyme activity.
Section 3.2: Elaborate on the potential mechanisms by which AFB1 disrupts DNA replication.
Consider adding a section addressing the limitations of the study and future research directions.
The overall language quality seems good, but a thorough proofreading for grammar and clarity is recommended.
This manuscript presents a well-conducted study with significant findings. Addressing the weaknesses mentioned above and incorporating the suggested edits will strengthen the manuscript.
Comments on the Quality of English LanguageThe overall language quality seems good, but a thorough proofreading for grammar and clarity is recommended.
Round 2
Reviewer 2 Report
Comments and Suggestions for Authors
The concerns raised in the previous feedback have been successfully resolved in the updated manuscript. Here is a brief summary of the changes made:
The strengths of the manuscript have been acknowledged. Issues regarding AFB1 degradation mechanisms, ELISA kit specificity, coumarin usage, incubation time, statistical analysis, controls, result presentation, discussion, and language clarity have all been addressed.
Comments on the Quality of English LanguageA few grammatical errors were spotted which can be corrected during galley proofreading if the manuscript is accepted.